# Enhanced Shielding Performance of Layered Carbon Fiber Composites Filled with Carbonyl Iron and Carbon Nanotubes in the Koch Curve Fractal Method

**DOI:** 10.3390/molecules25040969

**Published:** 2020-02-21

**Authors:** Hetong Zhang, Yue Guo, Xiang Zhang, Xinqian Wang, Hang Wang, Chunsheng Shi, Fang He

**Affiliations:** 1School of Materials Science and Engineering and Tianjin Key Laboratory of Composite and Functional Materials, Tianjin University, Tianjin 300072, China; zhanght@tju.edu.cn (H.Z.); yueguo@tju.edu.cn (Y.G.); zhangxiang@tju.edu.cn (X.Z.); xqwangtju@163.com (X.W.); 2117208019@tju.edu.cn (H.W.); csshi@tju.edu.cn (C.S.); 2Key Laboratory of Advanced Ceramics and Machining Technology, Ministry of Education, Tianjin University, Tianjin 300072, China

**Keywords:** carbon fiber composites, koch curve fractal, shielding effectiveness, carbon nanotubes, carbonyl iron powders

## Abstract

Layered carbon fiber composites (CFC) with enhanced shielding effectiveness (SE) were prepared with mixed fillers of carbon nanotubes (CNTs) and carbonyl iron powders (CIPs) in the form of a Koch curve fractal. In the layered composite structure, glass fiber (GF) cloth was used in the wave–transmissive layer (WTL), and the carbon fiber (CF) cloth was used in the supporting layer (SL). Between WTL and SL, CNTs and CIPs were distributed in epoxy resin in the form of a Koch curve fractal to act as an absorbing layer (AL), and copper foil was used as a reflective layer (RL) and bonded at the bottom of the whole composites. The layered structure design and excellent interlayer interface integration obviously improved the SE performance of the CFC. The SE of different samples was investigated, and the results show that, with the increase in the number (n) of Koch curve fractals, the SE of the samples enhanced in the low frequency scope (1–5 GHz). The sample with n = 2 has the highest SE value of 73.8 dB at 2.3 GHz. The shielding performance of the fractal sample filled by CNTs and CIPs simultaneously has a comprehensive improvement in the whole scope of 1–18 GHz, especially for the sample with n = 2. The cumulative bandwidth value of the SE exceeding 55 dB is about 14.3 GHz, accounting for 85% of the whole frequency scope, indicating the composite fabricated in this paper is an electromagnetic shielding material with great prospect.

## 1. Introduction

Electromagnetic interference (EMI) shielding materials are widely used in the military, industry, aerospace, and electronics to reduce electromagnetic interference [1]. Carbon fiber composite (CFC), a kind of EMI shield with high shielding effectiveness (SE), is believed to be a substitute for conventional metal-based shields because of its low density, high strength, and high resistance to corrosion [2]. However, its insufficient electrical conductivity limits the improvement efficiency of the SE for CFCs [3]. To resolve this problem, various solutions have been proposed and researched extensively. Among them, fabricating multilayer structured composites with electromagnetic protection packing between layers has attracted great attention due to their high SE and feasible designability [4,5,6].

Carbonyl iron powders (CIPs) are a kind of traditional magnetic loss SE filler with good temperature stability and high magnetic conductivity [7]. However, due to their small size, large specific surface areas, and low dielectric constants, CIPs did not show obvious advantages in EMI shielding materials. Moreover, CIPs cannot be used in large quantities because of their intrinsic high density. Therefore, it is more reasonable for CIPs to be used together with lightweight and high conductivity electromagnetic protection packing [8].

As a new type of carbon nanofiller, carbon nanotubes (CNTs) are widely used to improve the electromagnetic protection properties of CFCs due to their high conductivity and aspect ratio [9]. However, CNTs are not suitable to be used on a large scale because of their high cost. Many studies have shown that patterning of loss materials can have an excellent protection effect on electromagnetic waves in multiple frequency scope [10]. Xia et al. [11] studied the effectiveness of the Koch fractal theory on the electromagnetic wave absorbing effect of the activated carbon fiber felt. The results show that the fractal number and fractal pattern distribution have obvious effects on enhancing electromagnetic protection effect and widening frequency scope. At present, most research results fail to solve the problems of narrow electromagnetic shielding frequency scope and poor shielding strength [12,13,14]. However, fractal pattern design can add different kinds of electromagnetic protection fillers in different areas to form anisotropic composite materials, so as to increase the frequency scope of electromagnetic protection and broaden the bandwidth of electromagnetic protection. In addition, there is an urgent need in the field of electromagnetic shielding to improve the bandwidth and multi-frequency effects of fractal structures.

In this paper, the fractal layer composed of the carbonyl iron powder-filled epoxy (CIP@EP) and the carbon nanotubes-filled epoxy (CNT@EP) was used as an absorbing layer (AL) in the form of a Koch Curve fractal. At the same time, in the CFC, carbon fiber (CF)-filled epoxy was used as a supporting layer (SL), glass fiber (GF)-filled epoxy was used as a wave–transmissive layer (WTL), and copper foil on the bottom was used as a reflective layer (RL). The layered structure CFC with Koch curve fractal design prepared in this paper shows enhanced EMI shielding performance and a multi-frequency protection effect, which will be one kind of good EMI shielding material.

## 2. Results and Discussion

Figure 1 shows the morphology and basic parameters of CNTs and CIPs fillers. As shown in Figure 1a,b, the particle size of CIPs is about 1 μm, and there are some bumps on the surface of the CIPs particle at about 50 nm. Figure 1c shows that the saturation magnetization (Ms) and coercive force (Hc) of CIPs are 152 emu/g and 30 Oe, respectively, which makes CIPs obtain relatively high magnetic loss of electromagnetic waves [15,16]. The SEM and TEM pictures of CNTs using in this paper are shown in Figure 1d,e. It can be seen that CNTs are 20 nm in diameter and below 10μm in length, and as demonstrated in the HRTEM image in Figure 1e, the wall thickness of CNTs is about 12 layers. The CNTs show clear multi-walled structure and excellent crystallinity, which is also proved by the Raman test result from Figure 1f. The ID/IG ratio of CNTs is 0.614, indicating that CNTs have good crystallinity, which gives CNTs high conductivity and strong electrical losses to electromagnetic waves [17,18]. 

As we all know, the electromagnetic shielding mechanism of materials can be divided into reflection, multiple reflection, and absorption. Therefore, when fabricating electromagnetic shielding materials, the components in composite materials such as reinforcement, matrix, and their interface should be fully utilized to maximize their electromagnetic shielding effect. At the same time, the structure design is also a key factor for the composites used as electromagnetic shielding. In this paper, a layered composite material was designed and prepared, in which the Koch curve fractal was adopted in the AL, and CNTs and CIPs were filled inside and outside the fractal curve parts, respectively, as shown in Figure 2a–f. In order to prevent the CNTs and CIPs in the AL from being mixed with each other during the preparation process, a two-step curing method was adopted to fabricate AL in the samples with fractal numbers of zero, one, and two. From Figure 2b,d,f, it can be seen that the boundary lines between CNTs and CIPs-filled areas are clear and neat, and Koch fractal design of AL is well realized. Then, a GF layer with good impedance matching with air as WTL was fabricated on top of the AL, and copper foil as electromagnetic wave RL was placed to the bottom of the composites. After that, hot press molding process was used to prepare the final layered carbon fiber electromagnetic shielding composite. The cross section of the composites was observed through SEM, as shown in Figure 2g,h. It can be seen that the composite material has a clear, layered structure. The copper foil is tightly bonded to the composite material, and the thicknesses of the SL, AL and WTL are 1, 0.7 and 1.16 mm, respectively. According to the literature research and the characteristics of the materials, GF was used as WTL to enable more electromagnetic waves to enter the interior of the composite because of their good impedance matching with the air, and CFs acted as SLs, which also partially reflected electromagnetic waves due to their remarkable mechanical properties and conductive properties. CIPs and CNTs, representatives of traditional metal absorbers and new carbon nanomaterial absorbers, have magnetic and electrical loss effects on electromagnetic waves, respectively. The CIP@EP and the CNT@EP were used as AL to reduce electromagnetic wave reflection as much as possible. The copper foil as the RL can increase multiple reflection and absorption inside the composite material, thereby improving its overall shielding performance [19,20]. 

As is well known, filler distribution and the interface between fillers and the matrix in composite materials are two key factors for obtaining their good performance [21]. Therefore, in order to characterize the filler distribution and interface in the composites prepared by the process mentioned above, natural fractures of the composites obtained after liquid nitrogen cooling were characterized by SEM and were shown in Figure 3. It can be seen from Figure 3a,b that there is no obvious aggregation of CFs in the SL, and some EP fragments bond on the CFs surface, indicating that CFs are tightly bonded to EP and CFs play an excellent structural support role for the composites [22]. As shown in Figure 3c,d, the GFs in the WTL are uniformly distributed in EP and have good adhesion to EP, which is similar to the CFs in SL. Moreover, after observing the area where the CNTs are concentrated in the AL, it can be found that the CNTs are uniformly dispersed in the matrix, and can be effectively overlapped to form a conductive network [23], as shown in Figure 3e. From Figure 3f, it can be seen that the CIPs particles are also uniformly dispersed in the matrix, which ensures the magnetic loss effect of the CIPs [24]. 

According to the Electromagnetic Shielding Material Shielding Effectiveness Measurement Method (GJB 8820-2015), the performance of the samples should be tested in a shielded dark room [25]. The microwave measurement-specific framework for SE evaluation is a window test in free space. As shown in Figure 4a,b, the sample was tested in the shielded dark room. The measurement configuration diagram of the shielded chamber method is shown in Figure 4c [26]. 

A vectorial network analyser (VNA) is employed to measure the scattering parameters among the four antennas: one transmitting antenna in each chamber and one receiving antenna in each chamber.

In particular, the SE of the walls is measured for normal incidence by comparing the module of the scattering parameters of transmission S_21_ (dB) in free space with that obtained when the wall is placed between the antennas.
SE(dB) = S_21_ (free space) − S_21_ (material).(1)

The main problem in these measurements is to preserve a constant distance and a correct alignment between antennas. Further, the errors due to multiple paths caused by the boundaries of the wall under test and by other walls and objects in close proximity to the antennas must be reduced as much as possible [27,28,29]. 

The calculation formulas of shielding effectiveness are as follows: (2)$$\mathrm{SE}\;=\;20\log\frac{P_{0}}{P_{1}}$$
SE—Shielding effectiveness, the unit is decibel (dB).P_0_—Power received without shielding material, the unit is Watt (W).P_1_—Power received with shielding material, the unit is Watt (W).

Electromagnetic shielding tests were performed on different samples, and the results are shown in Figure 5. The properties of six samples were tested and shown in the manuscript. It can be seen from Figure 5a that the sample with n = 2 possesses an excellent electromagnetic shielding performance compared to the control sample, and its SE at 1–18 GHz is mostly more than 55 dB. The SE-frequency curve of the sample with n = 2 shows multiple shielding peaks. Figure 5b compares the SE of the samples with different fractal numbers. It can be seen that as the number of fractal numbers increases, the enhanced SE of the composites is obtained, especially at a frequency scope of 1–5 GHz and 15–18 GHz. In order to evaluate the synergistic effect of CNTs and CIPs in Koch curve fractal design on the SE of the composites with n = 2, the SE-frequency curves of the samples filled with CNTs plus CIPs, CNTs, and CIPs respectively are shown in Figure 5c. It is obvious that the sample with only CNTs has significant shielding peaks at 5–6 GHz and 13–17 GHz, and the one filled with CIPs shows high SE at frequency scope of 5–6 GHz. While the sample filled with both CNTs and CIPs exhibits higher SE at almost the whole frequency of 1–18GHz. The cumulative bandwidth values of the SE exceeding 55 dB in the shielding curves of different samples are compared in Figure 5d. It can be seen that the cumulative bandwidths of the samples filled with CNTs and CIPs simultaneously are all higher than those of the samples filled with only CNTs and only CIPs. For example, the cumulative bandwidth of the sample with n = 2 is about 14.3 GHz, accounting for 85% of the whole frequency scope, while those of the samples filled with only CNTs and only CIPs were 3.1 GHz and 6.5 GHz, respectively, indicating that CNTs and CIPs in the composites with Koch curve fractal design have an obviously synergistic effect on the enhanced electromagnetic shielding property. 

The samples in this paper showed excellent electromagnetic shielding performance, the shielding mechanism of which was analyzed to establish an electromagnetic model diagram as shown in Figure 6a,b. When an electromagnetic wave irradiated the surface of the composite, the WTL filled with GFs enabled more electromagnetic waves to enter the interior of the composite because of the good impedance of the GFs matching with the air. Then the electromagnetic waves passing through WTL were absorbed by CNTs and CIPs distributed in AL in electrical loss and magnetic loss. And the structural self-similarity of the Koch curve fractal in AL enabled the multi-scale electromagnetic protection of electromagnetic waves, resulting in multi-frequency shielding effects. Also, CNTs and CIPs formed more interfaces with epoxy in AL, respectively, causing electromagnetic waves to scatter and reflect. When the remaining electromagnetic waves entered the SL after passing through the AL, CFs also caused electromagnetic wave loss through electrical loss and interface loss. There were three ALs and SLs in the composites fabricated in the paper, which caused more electromagnetic waves to be absorbed and lost by these ALs and SLs. After that, a few electromagnetic waves reached the RL where the copper foil was located, and then were reflected and re-entered into the composite to achieve secondary or even multiple losses. Moreover, at the interface junction between the layers, the electromagnetic waves were also lost through multiple reflections due to changes in impedance. Eventually, the shielding effectiveness of the composite can be maximized through such electromagnetic wave propagation and dissipation paths.

## 3. Experimental Procedures

### 3.1. Raw Materials

CNTs and CIPs, fillers in the AL, were provided by Shanghai Macklin Biochemical Technology Co. Epoxy resin (E-51) and triethylenetetramine purchased from the Tianjin Institute of Synthetic Materials were used as raw materials of the matrix in the AL. WTL and SL were fabricated by CF prepreg and GF prepreg purchased from Weihai Guangwei Composite Material Co., Ltd. (Weihai, China) and their properties were listed in Table 1. 

The properties of CF prepreg and GF prepreg were acquired from the product manual of Weihai Guangwei Composite Material Co., Ltd. The CF (WP3021) represents the product model of CF prepreg. The GF(218#) represents the product model of glass fiber prepreg. The 2/2 twill indicates that the fibers are woven in a 2 press 2 twill (45°) textured prepreg woven fabric. Other properties include the main performance parameters of conventional prepreg. The mechanical properties and shielding properties of the samples will be affected by the different weaving methods of the fibers. The research results show that the CF-woven with 45° twill has better mechanical properties and shielding effects, so we chose the CF prepreg and GF prepreg of this type. 

### 3.2. Characterization

The shielding effectiveness was measured with the vector network analyzer (Agilent W449001, Agilent Technologies Inc. Santa Clara, CA, USA) in AECC Beijing Institute of Aeronautical Materials. Scanning electron microscopy (SEM, equipment type: PHILIPS XL30) was applied to observe the surface morphologies of CFC and the fracture surfaces of different samples at an operating voltage of 10 kV. All samples were sputtered with Au coating to improve the electrical conductivity prior to SEM observation. The microstructures of CNTs were investigated by the transmission electron microscopy (TEM, JEOL JEM-2100F, Tokyo, Japan). The saturation magnetization (Ms) and coercive force (Hc) of CIPs were evaluated on a vibrating sample magnetometer (VSM, JDM-13, Quantum Design China subsidiary, Beijing, China) and the field reached up to 1.5 × 104 Oe. Raman spectroscopy (Renishaw inVia Raman Microscope, Renishaw, Gloucestershire, London, UK) with 532 nm Ar+ laser was used to analyze the structure of CNTs. 

### 3.3. Preparation

Before preparing the samples, CF prepreg and GF prepreg were both cut into square pieces with a side length of 180 mm. Firstly, a piece of CF prepreg was placed in the mold, and then one layer of fractal AL was loaded on the CF prepreg through two-stepped curing method. In brief, a specific fractal template was placed on the CF prepreg, and then a uniform mixture of CIP, EP resin, and curing agent was poured into the mold. The area without the template was filled by the mixture. Then the mold was kept at 60 °C for 30 min in a vacuum oven to make the mixture a semi-cured state. Then, after removing the fractal template, a uniform mixture of CNTs, EP resin, and curing agent was put into the place where the template was removed and kept at 60 °C for 30 min to form CNTs@EP area of AL. In the fractal areas, weight ratios of CIPs and CNTs were kept at 30% and 5%, respectively. In this process, the thickness of the fractal template was 0.7 mm and the mass ratio of EP resin and triethylenetetramine was 10:1. By the same method mentioned above, three layers combined with AL and CF prepreg were obtained, and then one piece of GF prepreg was put on the top AL and one piece of copper foil was placed under the lowest layer of CF prepreg. At last, all the layers in the mold were cured in a vacuum oven at 120 °C for 2 h to obtain CFC samples with the whole thickness of 6 mm, and a sketch of its structure is shown as Figure 5b. A control sample without AL filled with CIPs and CNTs and the samples with AL filled by only CNTs or CIPs were also prepared for comparison purpose, and represented by control, CNT, and CIP. 

## 4. Conclusions

The SE performance of the composite material was improved significantly by the layered structure design of GF, CF and copper foil combined with fractal layer. Compared with the control sample without CNT@EP and CIP@EP, the performance was improved in all aspects. The shielding performance of the composite had multiple shielding peaks in frequency range of 1–18 GHz due to the multiple loss mechanism of CNT@EP and CIP@EP in the Koch curve fractal. As a result, it had a more balanced shielding effect in the whole frequency scope compared with the samples containing only one type of filler. As the number of fractals increased, the shielding effect at 1–6 GHz shifted towards the low-frequency direction and increased continuously, which was confirmed as a special shielding effect brought by the structural self-similarity of the Koch fractal. 

## Figures and Tables

**Figure 1 molecules-25-00969-f001:**
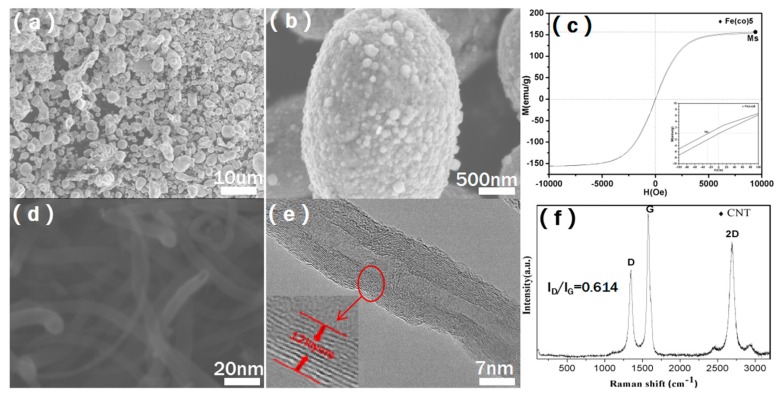
SEM images (**a**,**b**) of the carbonyl iron powders (CIPs); VSM image (**c**) of the CIPs; SEM images (**d**,**e**) of the carbon nanotubes (CNTs); the Raman image (**f**) of the CNTs.

**Figure 2 molecules-25-00969-f002:**
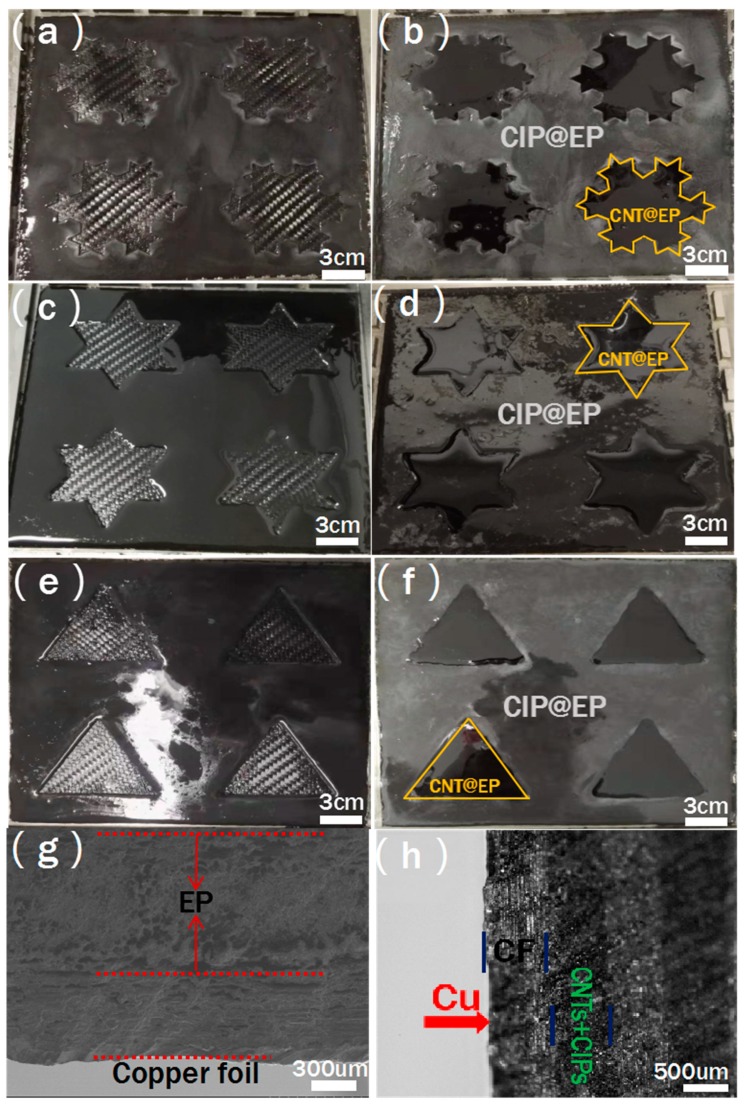
Pictures of absorbing layer (AL) in the samples with n = 2 (**a**,**b**); n = 1 (**c**,**d**); n = 0 (**e**,**f**); SEM (**g**); and optical (**h**) microscopes of the cross section for the sample with n = 2.

**Figure 3 molecules-25-00969-f003:**
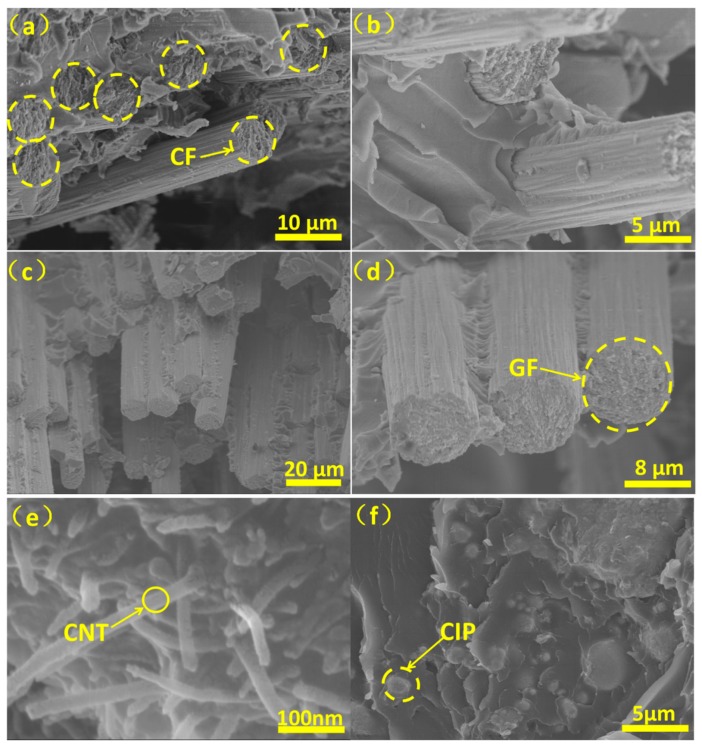
SEM fractures images of SL (**a**,**b**), WTL (**c**,**d**) and AL (**e**,**f**) of the samples with n = 2.

**Figure 4 molecules-25-00969-f004:**
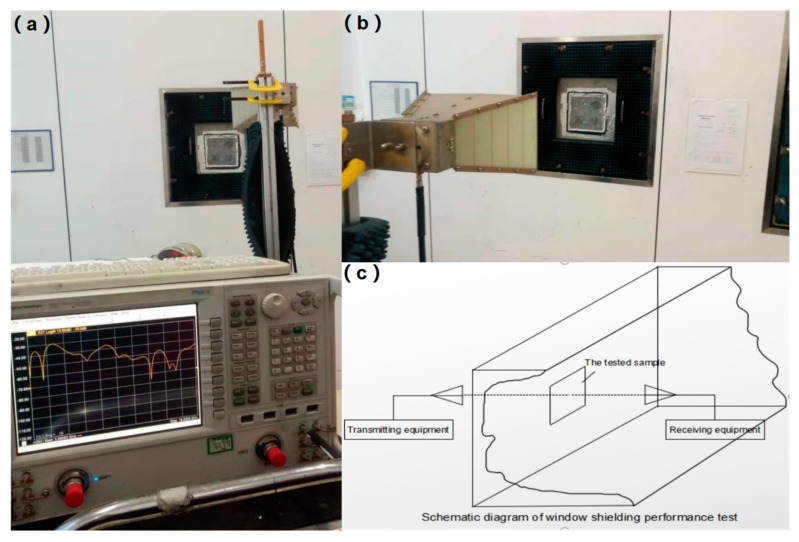
Physical pictures of shielded chamber measurement (**a**,**b**), and measurement configuration diagram of shielded chamber method (**c**).

**Figure 5 molecules-25-00969-f005:**
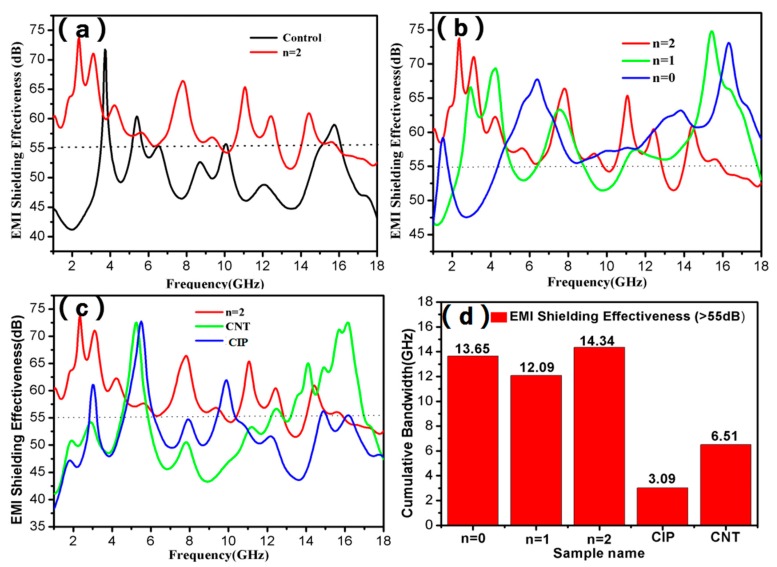
SE–frequency curve comparison of sample with n = 2 and control (**a**), samples with different fractal number (**b**), samples with different fillers (**c**), and cumulative bandwidth values of the SE exceeding 55 dB for all samples (**d**).

**Figure 6 molecules-25-00969-f006:**
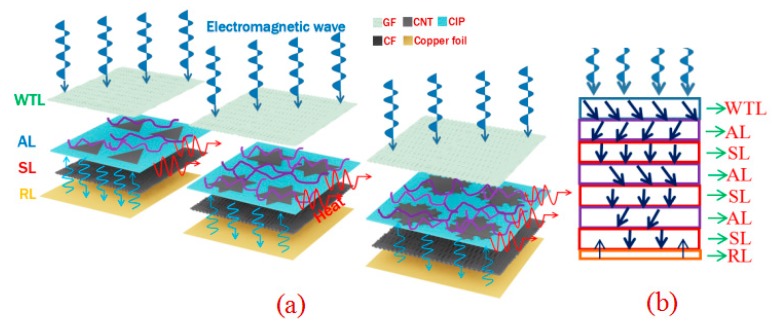
Schematic diagram of shielding mechanism (**a**) main perspective (**b**), sectional view.

**Table 1 molecules-25-00969-t001:** Properties of glass fiber (GF) prepreg and carbon fiber (CF) prepreg.

Fiber (Brand)	Method of Knitting	0° Tensile Strength (MPa)	Fiber Areal Weight (g/m^2^)	Thickness (mm)	Resin Volume Content (%)
CF(WP3021)	2/2 twill	670	198	0.25	40
GF(218#)	2/2 twill	526	220	0.29	35 ± 5

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
