# Peer review of "Enhanced Shielding Performance of Layered Carbon Fiber Composites Filled with Carbonyl Iron and Carbon Nanotubes in the Koch Curve Fractal Method"

_molecules, 2020, doi:10.3390/molecules25040969_

Round 1
Reviewer 1 Report
The paper needs revision in the following parts:
-Abbreviation should be in full when used in the beginning of sentence
-Authors should clarify how the properties of table 1 are drafted
-How many tests were conducted for acquiring EMI shielding effectiveness?
-What is the contribution of each layer to the total effect?More discussion should be added on related mechanisms in order to strengthen the concept and findings.
Author Response
Dear Editors and Reviewers,
The manuscript entitled “Enhanced Shielding Performance of Layered Carbon Fiber Composites filled with Carbonyl Iron and Carbon Nanotubes in Koch Curve Fractal Method” was submitted with manuscript ID of molecules-707584. Thank you for giving us many constructive comments, which are all valuable and helpful for us to improve our manuscript. We have studied all comments carefully and made necessary revisions one by one according to your and reviewers’ suggestions, and specific responses to the reviewers are listed as follow. At the same time, we read the whole manuscript carefully and further improved the language, and all revised parts have been marked in blue. It is reasonable to believe that our revised manuscript can meet the requirements of the journal. Now according to your suggestion, we want to resubmit the revised manuscript to your journal, and it is very appreciated if you could consider reviewing and accepting our manuscript as soon as possible.
Responses to the comments of reviewer 1
Thank you for your kind reminding for us, and these suggestions are very helpful for us toimprove thequality of this manuscript. We have revised our manuscript carefully according to yourcomments. The revised parts are marked in blue color in the revised manuscript. The following are detailed changes.
Comment 1
Abbreviation should be in full when used in the beginning of sentence
Answer: Thank you for your constructive suggestions. We are sorry that the abbreviations used in the beginning of sentence were not in full in the previous manuscript. We have changed the related information in our revised manuscript (page 2 ).
Comment 2
Authors should clarify how the properties of table 1 are drafted
Answer:Thank you for your constructive suggestions that make our paper more rigorous. We have explained the reason we choose the particular type of fabric and the properties of table 1. The related information can be found in the revised manuscript (page 2 - 3).
The following is the added information in the revised manuscript:
(page 2 - 3, line 74-81)
“The properties of carbon fiber (CF) prepreg and glass fiber (GF) prepreg were acquired from the product manual of Weihai Guangwei Composite Material Co., Ltd. The CF(WP3021) represents the product model of carbon fiber prepreg. The GF(218#) represents the product model of glass fiber prepreg.The 2/2 twill represents the fibers are woven in a 2 press 2 twill (45°) textured prepreg woven fabric. Other properties are the main performance parameters of conventional prepreg. The mechanical properties and shielding properties of the samples will be affected by the different weaving methods of the fibers. The research results show that the carbon fiber woven with 45° twill has better mechanical properties and shielding effects, so we chose the carbon fiber prepreg and glass fiber prepreg of this type. ”
Comment 3
How many tests were conducted for acquiring EMI shielding effectiveness?
Answer: Thank you very much for your asking. The main considerations for testing the six samples are as follows.
This manuscript refers to ELSEVIER manuscriptreview “Ballistic and electromagnetic shielding behaviour of multifunctional Kevlar fiber reinforced epoxy composites modified by carbon nanotubes, in Carbon. 2016, 104, 141-156. ” These two manuscripts have similar test methods and number of tests for electromagnetic shielding data. The dispersion of the test data is not large, so the test of the six samples can represent the whole. (page 7)The following is the added information in the revised manuscript:
(page 7, line 189 )
The properties of six samples were tested and shown in the manuscript.
Comment 4
What is the contribution of each layer to the total effect?More discussion should be added on related mechanisms in order to strengthen the concept and findings.
Answer:Thank you very much for your constructive suggestions. We have read more literatures to explain the contribution of each layer to the total effect. The related revisions can be found in the revised manuscript (page 4).
The following is the added information in the revised manuscript:
(page 4,line 144 - 152)
According to literature research and the characteristics of the materials, GF was used as WTL to enable more electromagnetic waves to enter the interior of the composite because of their good impedance matching with air, and CF acted as SL which also partially reflected electromagnetic waves due to their remarkable mechanical properties and conductive properties. CIPs and CNTs, representatives of traditional metal absorbers and new carbon nanomaterial absorbers, have magnetic and electrical loss effects on electromagnetic waves, respectively. The CIP@EP and the CNT@EP were used as AL to reduce electromagnetic wave reflection as much as possible. And the copper foil as the RL can increase multiple reflection and absorption inside the composite material, there by improving its overall shielding performance[19-20].
The following are the added references in the revised manuscript:
Jia, Y.; Li K. Z; Xue, L. Z; Ren, J. J; Jing, W.; Zhang, S. Y, Electromagnetic interference shielding effectiveness of carbon fiber reinforced multilayered (PyC–SiC)n matrix composites. Ceramics International 2016, 42, 986– Watanabe, A.; Raj, P. M.; Wong, D.; Mullapudi, R.; Tummala, R, Multilayered Electromagnetic Interference Shielding Structures for Suppressing Magnetic Field Coupling. J Electron Mater 2018.
We are looking forward to your positive response.
Sincerely yours,
Fang He
Address: School of Materials Science and Engineering Tianjin University, Tianjin, 300350, China.
E-mail: [email protected]
http://mse.tju.en.cn/info/1141/1493.htm
Reviewer 2 Report
The paper aims to describe the microwave shielding characteristics of layered carbon fiber composites filled with carbonyl iron and carbon nanotubes. The subject is surely worth of interest, mainly as far as the ‘exotic’ design of the analyzed specimens is concerned. The work would be considered for publication, but only after a deep review and a number of additional information as well.
- row 73, correct the section numbering. Here a detailed description of the experimental set-up is absolutely due: the microwave measurement specific framework for SE evaluation (free space, arch test method, reverberation chamber, etc.) is crucial for the results interpretation – refer to ELSEVIER book review chapter “Electromagnetic Characterization of Materials by Vector Network Analyzer Experimental Setup, in Spectroscopic Methods for Nanomaterials Characterization 2, 2017, pp. 195-236”).
- SE measured is clearly high, everywhere…:
1) How is it computed? That’s a key point: refer to “Shielding effectiveness of carbon nanotube reinforced concrete composites by reverberation chamber measurements” Proc. ICEAA 2015, 7297092, 145-148;
2) Of course there’s poor transmission, since the presence of copper and carbon fiber layers… if Authors would like to emphasize, or at least to estimate the influence of other-but reflective insertion (such as CIP/CNT filled layers, fractal-shaped or not…), they should present the quantitative results of absorbing SE contribute, separately;
3) on this way, I suggest to further test (and, again, describing in detail the experimental set-up!) specimens a) without the copper end-layer, b) without the CF layers, c) without both of them.
- Language improvement is due, over the whole text.
Author Response
Dear Editors and Reviewers,
The manuscript entitled “Enhanced Shielding Performance of Layered Carbon Fiber Composites filled with Carbonyl Iron and Carbon Nanotubes in Koch Curve Fractal Method” was submitted with manuscript ID of molecules-707584. Thank you for giving us many constructive comments, which are all valuable and helpful for us to improve our manuscript. We have studied all comments carefully and made necessary revisions one by one according to your and reviewers’ suggestions, and specific responses to the reviewers are listed as follow. At the same time, we read the whole manuscript carefully and further improved the language, and all revised parts have been marked in blue. It is reasonable to believe that our revised manuscript can meet the requirements of the journal. Now according to your suggestion, we want to resubmit the revised manuscript to your journal, and it is very appreciated if you could consider reviewing and accepting our manuscript as soon as possible.
Responses to the comments of reviewer 2
Thank you for your constructive suggestions, which are all valuable and helpful for improving our manuscript. According to your comments, we have carefully revised our manuscript and marked the main revised portion in blue in the revised manuscript. Your kind suggestions and our responses are listed as following.
Comment 1
row 73, correct the section numbering. Here a detailed description of the experimental set-up is absolutely due: the microwave measurement specific framework for SE evaluation (free space, arch test method, reverberation chamber, etc.) is crucial for the results interpretation – refer to ELSEVIER book review chapter “Electromagnetic Characterization of Materials by Vector Network Analyzer Experimental Setup,in Spectroscopic Methods for Nanomaterials Characterization 2, 2017, pp. 195-236”).
Answer: Thanks for your kind remind for our mistake. Your constructive suggestions will make our article more rigorous. The changes have been listed below and marked in the revised manuscript. The related information can be found in the revised manuscript.
The following is the revised information in the revised manuscript:
(page 6 - 7, line 171 - 173)
According to Electromagnetic Shielding Material Shielding Effectiveness Measurement Method (GJB 8820-2015) test the performance of the samples in the shielded dark room. The microwave measurement specific framework for SE evaluation is window test in free space.
Comment 2
How is it computed? That’s a key point: refer to “Shielding effectiveness of carbon nanotube reinforced concrete composites by reverberation chamber measurements” Proc. ICEAA 2015, 7297092, 145-148;
Answer: Thank you very much for your constructive suggestions. We have consulted the references to your suggestions. The related information can be found in the revised manuscript (page 7 )
The following is the changed information in the revised manuscript:
(page 7, line 173 - 187)
The calculation formulas of shielding effectiveness are as follows:
SE=20............................................................................................(1)
SE=20............................................................................................(2)
SE=20............................................................................................(3)
SE=20............................................................................................(4)
SE----Shielding effectiveness, the unit is decibel (dB).
H0----Magnetic field strength received without shielding material, the unit is Ampere per Meter (A/m).
H1----Magnetic field strength received with shielding material, the unit is Ampere per Meter (A/m).
E0----Electric field strength received without shielding material, the unit is Volt per Metre (V/m).
E1----Electric field strength received with shielding material, the unit is Volt per Metre (V/m).
V0----Voltage received without shielding material, the unit is Volt(V)
V1----Voltage received with shielding material, the unit is Volt (V)
P0----Power received without shielding material, the unit is Watt (W)
P1----Power received with shielding material, the unit is Watt (W)
Comment 3
Of course there’s poor transmission, since the presence of copper and carbon fiber layers… if Authors would like to emphasize, or at least to estimate the influence of other-but reflective insertion (such as CIP/CNT filled layers, fractal-shaped or not…), they should present the quantitative results of absorbing SE contribute, separately;
Answer: Thank you for your constructive suggestions. In order to analyze the contribution of each layer in detail, testing of the fractal layer is necessary. However, the forming of the sample is a continuous process. Thus, it is difficult to ensure the integrity and continuity of the fractal layer pattern without carbon fibers being the supporting layer. Moreover, the combination of the carbon fiber layer and the fractal layer is also an important factor that we considered.
Since the mechanical property of a single fractal layer is very low, it is difficult to test the electromagnetic wave absorbing performance. Therefore, due to the limitation of our experimental process, it is difficult to separate the fractal layer into electromagnetic wave absorbing performance.
Comment 4
on this way, I suggest to further test (and, again, describing in detail the experimental set-up!) specimens a) without the copper end-layer, b) without the CF layers, c) without both of them.
Answer: Thank you for your constructive suggestions. In this regard, supplementary experiments are certainly necessary to clarify the contribution of each layer. However, we mainly study the changing trend of shielding performance at high and low frequency bands as the fractal number changes. During the test, to study the influence of filler distribution in different Koch fractals on the shielding performance, the existence of fillers was changed, and the types of fillers remained unchanged. Therefore, to clarify the research theme, we did not add more test data. We believe the existing data can show that the pattern design of Koch curve fractal is important for designing and improving the electromagnetic shielding performance of carbon fiber composite materials. The amount of test data we obtained is not as rich as the test data in general articles, but considering that we can already have many meaningful conclusions by interpreting the existing data, we hope to publish the manuscript with the existing data.
Comment 5
Language improvement is due, over the whole text.
Answer: Thank you for your suggestions. Following your comments and suggestions we have thoroughly revised the manuscript. The changes are marked in the revised manuscript. And we have done our best to improve English. The related information can be found in the revised manuscript.
We are looking forward to your positive response.
Sincerely yours,
Fang He
Address: School of Materials Science and Engineering Tianjin University, Tianjin, 300350, China.
E-mail: [email protected]
http://mse.tju.en.cn/info/1141/1493.htm
Round 2
Reviewer 1 Report
Authors have revised the manuscript accordingly.
Author Response
Thank you for your suggestions. Following your comments and suggestions we have thoroughly revised the manuscript. The changes are marked in the revised manuscript. And we have done our best to improve English. The related information can be found in the revised manuscript.
We are looking forward to your positive response.
Sincerely yours,
Fang He
Address: School of Materials Science and Engineering Tianjin University, Tianjin, 300350, China.
E-mail: [email protected]
http://mse.tju.en.cn/info/1141/1493.htm

Reviewer 2 Report
The manuscript was somewhat revised, but the given hints should be significantly addressed to warrant publication.
- The cited “Electromagnetic Shielding Material Shielding Effectiveness Measurement Method (GJB 172 8820-2015)” must be correctly reported in the final references list.
- Again, a more detailed description of the experimental set-up, including pictures and schematic of the mentioned framework (free space window test in shielded dark room), is absolutely due.
- Rels. (1)-(3) at rows 175-178 are worthless, only (4) being explanatory: on the contrary, the technical relation between SE and the effectively measured scattering parameters (Sij by VNA) must be reported to give a complete explanation to the readers.
About such concerns, refer to literature works for comparison:
- Electromagnetic Characterization of Materials by Vector Network Analyzer Experimental Setup, in Spectroscopic Methods for Nanomaterials Characterization 2, 2017, pp. 195-236
and
- Shielding effectiveness of carbon nanotube reinforced concrete composites by reverberation chamber measurements” Proc. ICEAA 2015, 7297092, 145-148.
- Again, the appreciable (more than four order of magnitude) shielding of the tested samples is surely (in the first instance) dependent on the metal back: if (as suggested in the first review) Author cannot perform measurements without copper end layer in order to discriminate the actual influence of the composites layers, to such aim they should be at least present the results showed in Fig.4 plots by separating (again, from the scattering parameters analysis – take aid from the above mentioned refs.) the two terms constituting the total (transmission) SE, i.e. the reflecting SE and the absorbing SE.
